# Room-Temperature Hydrogen-Sensitive Pt-SnO₂ Composite Nanoceramics: Contrasting Roles of Pt Nano-Catalysts Loaded via Two Different Methods

Jieting Zhao [1,2], Jiannan Song [2], Xilai Lu [2], Menghan Wu [2], Zhiqiao Yan [3], Feng Chen [3] and Wanping Chen [2,*]

1   Engineering Research Centre for Deep Processing of Rare Metals, School of Engineering, Changchun Normal University, Changchun 130032, China; zhaojieting@ccsfu.edu.cn
2   School of Physics and Technology, Wuhan University, Wuhan 430072, China
3   Guangdong Provincial Key Laboratory of Metal Toughening Technology and Application, Institute of New Materials, Guangdong Academy of Sciences, Guangzhou 510650, China
*   Correspondence: wpchen@whu.edu.cn

**Abstract:** Soluble noble metal salts are widely used for loading noble metals as nano-catalysts in many applications. In this paper, Pt-SnO₂ composite nanoceramics were prepared from SnO₂ nanoparticles and H₂PtCl₆ using two Pt loading methods separately: for the solution reduction method, a H₂PtCl₆ solution was added to a suspension of SnO₂ and zinc powder to form Pt on SnO₂ nanoparticles, and for the impregnation method, Pt was formed from H₂PtCl₆ in the course of sintering. Although a series of samples prepared using both Pt loading methods showed a solid response to H₂ at room temperature, the ones prepared using the solution reduction method exhibited much better room-temperature hydrogen-sensing characteristics. For two samples of 0.5 wt% Pt and sintered at 825 °C, the response value for the sample prepared using the solution reduction method was 9700 to 1% H₂–20% O₂-N₂, which was much larger than the value of 145 for the sample prepared using the impregnation method. Samples prepared using the two Pt loading methods have similar microstructures characterized via XRD, FESEM, EDS, TEM, and HRTEM. However, the residual chlorine content in those using the impregnation method was higher than those using the solution reduction method according to the analysis. It is proposed that the striking difference in room-temperature hydrogen sensing characteristics among samples prepared using these two different Pt loading methods separately resulted from their different chlorine removal processes. This study demonstrates the importance of a proper method for loading noble metals from their soluble salts as nano-catalysts in many applications.

**Keywords:** SnO₂; gas sensor; room temperature; Pt loading; catalyst





## 1. Introduction

In recent years, gas sensors have been widely used in industrial production, environmental monitoring, healthcare, home safety, and many other fields [1–3]. Among various types of commercial gas sensors, the one based on metal oxide semiconductors (MOSs) is especially attractive due to its low cost, simple structure, and easy operation [4–6]. Commonly used MOS materials are classified as n-type (e.g., SnO₂, ZnO, WO₃, etc.) and p-type (CuO, NiO, Co₃O₄, etc.), of which SnO₂ has long been successfully commercialized as the most popular material [7]. Despite these advantages, commercial gas sensors based on SnO₂ thick films all have a major disadvantage of high operating temperatures (200–400 °C) to have strong responses to target gases, preventing them from some critical applications [8,9]. As an example, it is well known that hydrogen is used as a carbon-free energy carrier, with water as the only by-product. As such, it is widely used in the field of new energy sources. However, as hydrogen is flammable and explosive, commercial SnO₂ thick film hydrogen sensors operating at high temperatures have the risk of inducing hydrogen explosion

and have been prevented from the detection of $H_2$ in some important cases, such as in hydrogen energy vehicles [10]. Therefore, the development of low-temperature metal oxide semiconductor gas sensors has always been an essential goal in gas sensor research.

To develop MOS gas sensors with lower operation temperatures, researchers have modified MOS materials through changing material morphology [11], doping with metal ions [12], loading with noble metals [13], and/or constructing hetero-junctions [14]. Among these measures, noble metal loading has been proven to be especially effective in improving the gas-sensitive properties of MOSs. For example, Fan et al. showed a response value 2.5 times higher than that of pure $SnO_2$ to 100 ppm ethanol at 240 °C through loading 2 wt% Pt on pure $SnO_2$ nanosheets [15]. Wang et al. observed temperature-modulated double selectivity for CO at 130 °C and $CH_4$ at 240 °C through preparing Ag-loaded ZnO microspheres [16]. Zhu et al. demonstrated extraordinary gas sensitivity to CO at room temperature through preparing Pd-$SnO_2$ composite nanoceramics [17]. It is generally believed that catalytic noble metals can promote the decomposition of gas molecules, reduce the activation energy of gas adsorption and reactions, and thus decrease the operating temperature of MOS gas sensors.

Soluble noble metal salts (e.g., $H_2PtCl_6 \cdot 6H_2O$, $PdCl_2$, $HAuCl_4 \cdot 4H_2O$, $Pd(NH_3)_2(NO_2)_2$) are widely used to load noble metals on MOSs. In some research, MOSs were simply immersed in solutions of noble metal salts, and the loading of noble metals on MOSs was fulfilled through subsequent heat treatments [5,18–25], while in other research, some reducing agents (e.g., $NaBH_4$, $C_6H_8O_6$) [26–30] or metals (e.g., Zn) [17,31] were added to solutions of noble metal salts dispersed with MOSs to realize the loading of noble metals on MOSs through a solution reduction process, such as the loading of Pt through the addition of Zn [31]:

$$2Zn + Pt^{4+} \rightarrow Pt + 2Zn^{2+}, \tag{1}$$

Two entirely different methods have appeared for loading noble metals on MOSs in the literature, termed the impregnation method and the solution reduction method separately hereafter. Up to date, both methods have been extensively adopted to load noble metals on MOSs to improve their gas-sensing performance. At the same time, almost nobody has paid any attention to choosing between these two methods to achieve a better effect, or they are still determining if there is any difference between them.

In recent years, several advances have been made through investigating the room-temperature gas-sensitive properties of composite noble metal catalyst–metal oxide nanoceramics prepared via conventional pressing and sintering. Among them, Pt-$SnO_2$ composite nanoceramics have been well studied regarding their room temperature hydrogen sensing characteristics [13,31]. Room-temperature hydrogen sensors based on Pt-$SnO_2$ composite nanoceramics should be highly appealing for such applications as in hydrogen energy vehicles. In this study, Pt-$SnO_2$ composite nanoceramics were prepared with Pt loaded from chloroplatinic acid using two methods, namely the solution reduction method and the impregnation method, separately. While almost no differences could be observed between samples prepared using these two different Pt loading methods in their crystal structure or microstructure, a distinctive difference in the room-temperature hydrogen sensing capability could be easily observed: The ones prepared using the solution reduction method usually had much higher room-temperature hydrogen responses than those prepared using the other Pt loading method. For those samples prepared using the impregnation method, $Cl_2$ was formed from chloroplatinic acid at high temperatures and was proposed to be responsible for their observed lower room-temperature hydrogen responses. These results suggest that a high priority should be given to selecting a suitable method for loading noble metals on MOS to improve its gas sensing capability, which has been relatively neglected until now.

## 2. Results and Discussion

Though two different Pt loading methods were adopted in this study, there were no detectable differences in the X-ray diffraction patterns obtained for samples prepared using

these two methods separately. Four representative XRD patterns are shown in Figure 1, which were obtained for four samples of 2 wt% Pt, sintered at 600 °C and 825 °C, prepared using the solution reduction and impregnation methods separately. A heat treatment around 600 °C is common for the impregnation method and 825 °C has been found optimum for the sintering of room-temperature hydrogen-sensitive Pt-SnO$_2$ composite nanoceramics [13,31]. In all patterns, most strong peaks are identified as those of SnO$_2$ in a tetragonal rutile structure (JCPDS: 41-1445). In those of two samples sintered at 825 °C, some peaks from cubic Pt (JCPDS: 4-802) can be observed, while in those of two samples sintered at 600 °C, these peaks from Pt are much weaker or even cannot be detected. A similar result had been obtained in a previous study, whose intensity of Pt peaks also increased considerably with sintering temperature, and it was explained that the crystallinity of Pt nanoparticles increases with increasing sintering temperature [31]. Pt-SnO$_2$ bulk composites could be successfully prepared using these two Pt loading methods separately.

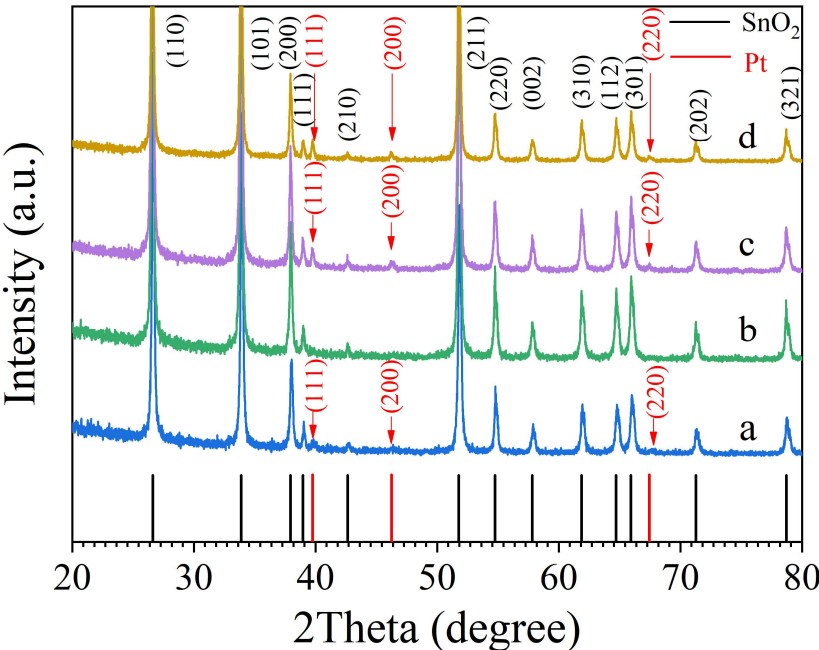

**Figure 1.** XRD patterns obtained for four samples of 2 wt% Pt: (a) using the solution reduction method for Pt loading and sintered at 600 °C; (b) using the impregnation method for Pt loading and sintered at 600 °C; (c) using the solution reduction method for Pt loading and sintered at 825 °C; (d) using the impregnation method for Pt loading and sintered at 825 °C.

Figure 2 displays some micro-structural analyses obtained for two samples of 2 wt% Pt and sintered at 825 °C while prepared separately using the solution reduction and impregnation methods for Pt loading. As shown in Figure 2a, a FESEM micrograph was taken for a fractured surface of a sample prepared using the solution reduction method. Most grains, which are expected to be SnO$_2$ grains, are around 50–70 nm and had experienced no obvious grain growth after sintering at 825 °C [17]. Micro-pores are presented in the microstructure, which are related to no detectable shrinkages in the sintering and supposed to be beneficial for gas sensing applications [31]. Figure 2b represents some EDS analyses obtained for this sample, in which a couple of Pt grains around 100 nm can be identified. This sample had a relatively high Pt content (2 wt%); other Pt nanoparticles were much smaller and could only be observed through TEM, which will be described in the following paragraphs. Figure 2c,d are micro-analyses obtained for another sample prepared using the impregnation method, which shows no detectable differences in micro-structure between samples prepared using two different Pt loading methods separately.

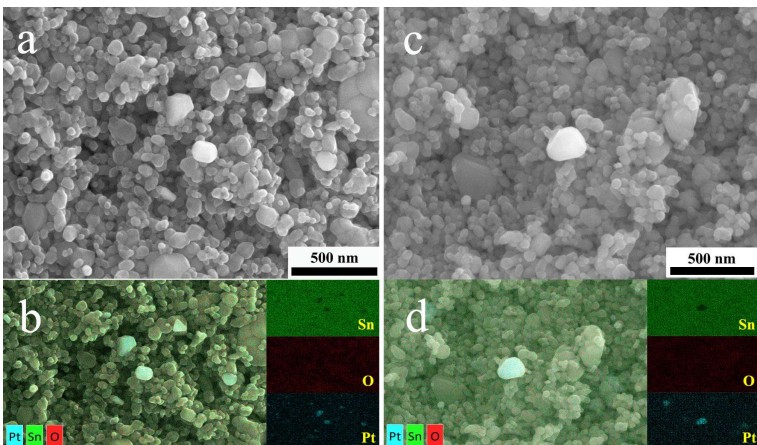

**Figure 2.** For samples of 2 wt% Pt and sintered at 825 °C: (**a**) FESEM micrograph for one prepared using solution reduction method; (**b**) EDS analyses for one prepared using solution reduction method; (**c**) FESEM micrograph for one prepared using impregnation method; (**d**) EDS analyses for one prepared using impregnation method.

These two samples in Figure 2 were partially crushed and ground into powders for further TEM analyses, and some of the results are shown in Figure 3. Those particles around 50–70 nm are $SnO_2$ grains. Additionally, smaller nanoparticles, typically as small as 5 nm, can be observed in both samples. HRTEM analysis was carried out for one such small nanoparticle in each sample, and the crystal plane spacing was calculated using Digital Micrograph software (Version 3.43.3213.0) to be 0.23 nm, corresponding to that of Pt (1 1 1). Therefore, these small nanoparticles were determined to be Pt nanoparticles, consistent with the XRD analysis. Samples prepared using these two different Pt loading methods were of quite similar microstructure even through TEM analysis.

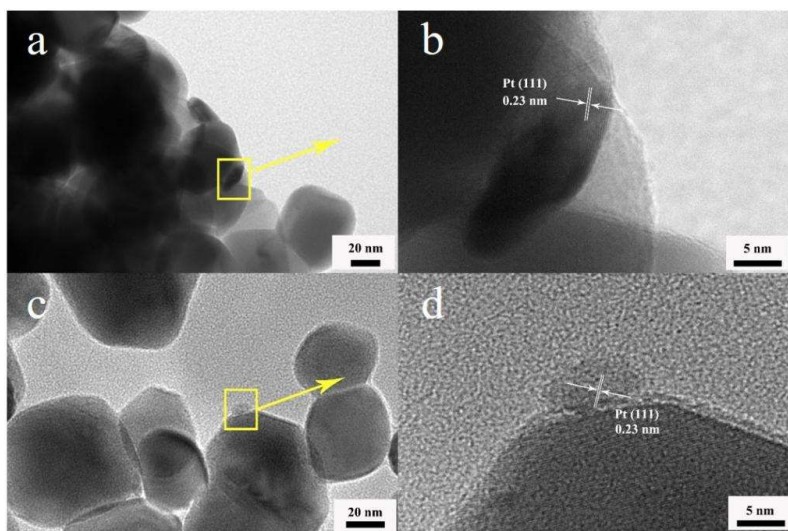

**Figure 3.** For samples of 2 wt% Pt and sintered at 825 °C: (**a**) TEM micrograph for one prepared using solution reduction method; (**b**) HRTEM of a small nanoparticle in one prepared using solution reduction method; (**c**) TEM micrograph for one prepared using impregnation method; (**d**) HRTEM of a small nanoparticle in one prepared using impregnation method.

For samples prepared using the impregnation method, sintering at an elevated temperature is crucial for achieving a room-temperature hydrogen sensing capability. In many investigations on loading noble metals on low-dimensional MOS materials to improve their gas sensing, a heat treatment is usually conducted around 600 °C after impregnation in

solutions of noble metal salts. For example, Matushko et al. prepared $SnO_2$ sensitive layers impregnated in a $H_2PtCl_6 \cdot 6H_2O$ solution and sintered them at 620 °C in air. The Pt-$SnO_2$ material showed an improved response/recovery time to $H_2$ at an operating temperature of 260 °C [5]. Yin et al. impregnated $SnO_2$ particles into a $HAuCl_4 \cdot 4H_2O$ solution and sintered them at 550 °C for 3 h in air to obtain Au-$SnO_2$ materials that could detect $H_2$ down to 0.4 ppm at an operating temperature of 350 °C [19]. Wang et al. impregnated $SnO_2$ nanosolid material into a $H_2PtCl_6 \cdot 6H_2O$ solution, sintered it at 500 °C for 2 h in $N_2$, and achieved a response value of 64.5 for 100 ppm CO at room temperature [18]. Ma et al. impregnated $SnO_2$ powder into a $Pd(NH_3)_2(NO_2)_2$ solution and sintered it at 500 °C in air. The resulting Pd-$SnO_2$ material inhibited poisoning by water vapor [21]. It is, therefore, meaningful to study those samples prepared using the impregnation method and sintered at 600 °C. For instance, the room-temperature hydrogen sensing characteristics of a sample of 0.5 wt% Pt, prepared using the impregnation method and sintered at 600 °C, has been carefully measured and shown in Figure 4.

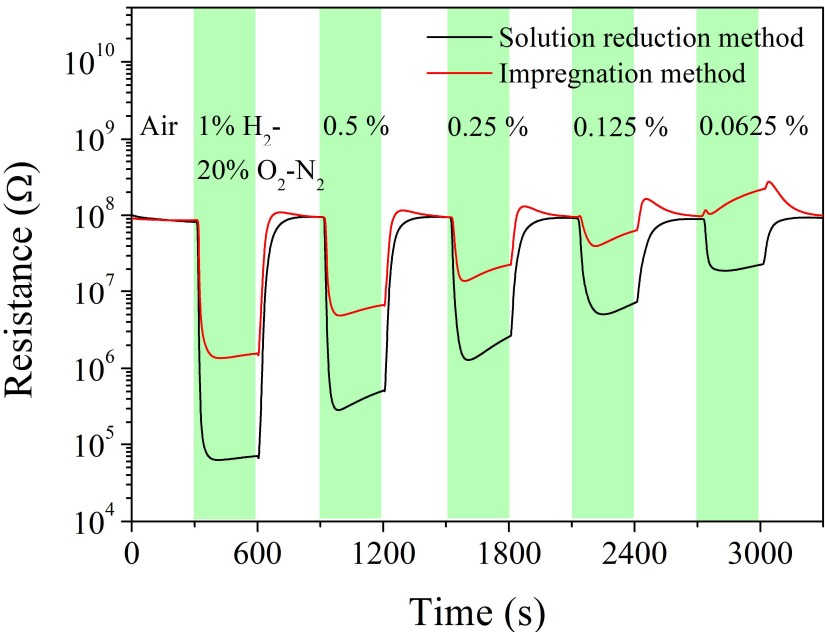

**Figure 4.** Room-temperature hydrogen sensing characteristics for two Pt-$SnO_2$ composite nanoceramic samples of 0.5 wt% Pt and sintered at 600 °C, prepared using the solution reduction and the impregnation methods for Pt loading, separately.

The response value of an n-type semiconductor gas sensor to a reducing gas is usually defined as $S = Ra/Rg$, where $Ra$ and $Rg$ are the resistances of the sensor in air and the target gas, respectively. As shown in Figure 4, the sample prepared using the impregnation method has a response of 65 to 1% $H_2$–20% $O_2$-$N_2$ at room temperature. Generally speaking, such a room-temperature response to $H_2$ is quite outstanding among those reported for low-dimensional MOSs in the literature [32–34]. However, as the $H_2$ concentration is reduced to 0.0625%, it gives no response.

Though Pt-$SnO_2$ composite nanoceramics prepared using these two different Pt loading methods were similar in microstructure, their room-temperature hydrogen sensing characteristics showed a quite different dependence on sintering temperature. As reported in a previous study, for samples prepared using the solution reduction method, an extraordinary room-temperature hydrogen sensing capability can be observed even before sintering [31]. For comparison, the room-temperature hydrogen sensing characteristics of a sample of 0.5 wt% Pt prepared using the solution reduction method and sintered at 600 °C is also shown in Figure 4. First of all, it can be seen that this sample has a response of 1300 to 1% $H_2$–20%$O_2$-$N_2$, which is quite typical for samples prepared using the solution reduction

method, while is 20 times higher than that of the sample prepared using the impregnation method [35,36]. Secondly, this sample has a response of 5 to 0.0625% $H_2$–20% $O_2$-$N_2$, which also contrast sharply with the other sample. Surprisingly, there is such a striking difference in room-temperature hydrogen-sensing characteristics between the samples prepared using the two different Pt loading methods separately.

For Pt-$SnO_2$ composite nanoceramics prepared using the solution reduction method, the optimum sintering temperature is around 825 °C, yielding good mechanical strength and appealing room-temperature hydrogen sensing characteristics [31]. Figure 5 shows the room-temperature hydrogen-sensitive properties of two samples of 0.5 wt% Pt sintered at 825 °C while prepared using these two different Pt loading methods separately to make a further comparison. To 0.0625%, 0.125%, 0.25%, 0.5%, and 1% $H_2$–20% $O_2$-$N_2$, the sample prepared using the impregnation method showed responses of 2.5, 8.5, 21, 53, and 145, respectively, as shown in the inset of Figure 5; the sample prepared using the solution reduction method showed responses of 20, 100, 477, 2190, and 9700, respectively. Meanwhile, the sample prepared using the solution reduction method also showed appealing response and recovery speeds. For 1% $H_2$–20%$O_2$-$N_2$, the response time and recovery time were 7 s and 160 s, respectively; both are highly outstanding for room-temperature hydrogen sensing. For each of the two Pt loading methods, the sample sintered at 825 °C showed a much better room-temperature hydrogen sensing capacity than the one sintered at 600 °C. Moreover, for this sintering temperature, the sample prepared using the solution reduction method still exhibited a much more attractive room-temperature hydrogen sensing capacity than the other one prepared using the impregnation method.

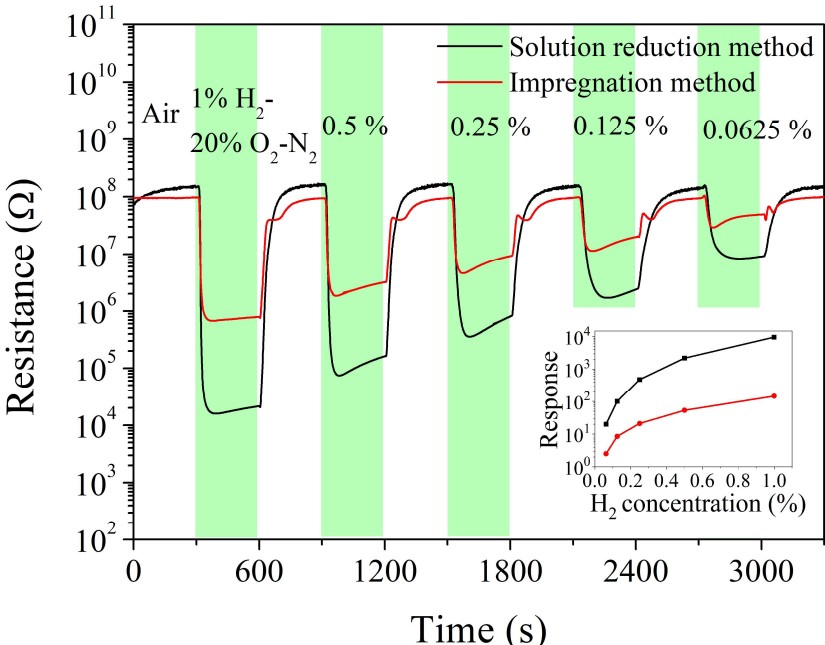

**Figure 5.** Room-temperature hydrogen sensing characteristics for two Pt-$SnO_2$ composite nanoceramic samples of 0.5 wt% Pt and sintered at 825 °C, prepared using the solution reduction and the impregnation methods for Pt loading separately. Inset: Response vs. $H_2$ concentration for the two samples.

Chlorine-containing precursors are frequently used to load noble metal nano-catalysts in many applications, while chlorine is generally known to have adverse effects on these nano-catalysts [37–39]. A post-treatment is mainly adopted to remove chlorine in the precursors effectively, and the thermal removal of chlorine from $H_2PtCl_6$ can be expressed explicitly as [37,40,41]:

$$H_2PtCl_6 \rightarrow PtCl_4 + 2HCl, \tag{2}$$

$$PtCl_4 \rightarrow Pt + 2Cl_2, \tag{3}$$

Obviously, for those samples prepared using the impregnation method in this study, chlorine must be removed in the course of sintering in this way. Table 1 shows the chlorine content of four samples of 2 wt% Pt prepared using the two different Pt loading methods and separately sintered at 600 °C and 825 °C.

**Table 1.** The content of residual chlorine in four samples of 2 wt% Pt prepared in this study.

| Sintering Temperature, °C | Pt Loading Method | Chlorine, wt% |
|---|---|---|
| 600 | Solution reduction | 0.167 |
| | Impregnation | 0.495 |
| 825 | Solution reduction | 0.14 |
| | Impregnation | 0.41 |

A small amount of chlorine can be detected in all four of these samples, and those prepared using the impregnation method had much more chlorine than those prepared using the solution reduction method. As a matter of fact, for those samples prepared using the solution reduction method, chlorine had been chiefly removed through a quite different process: as $(PtCl_6)^{2-}$ ions were dissolved in solution, chlorine was mostly removed from the samples through the centrifugation process. According to the results in Table 1, it is clear that chlorine can be more effectively removed through this method than thermal removal.

Given the fact that the amount of residual chlorine was considerably higher in samples prepared using the impregnation method than those prepared using the other method, a specially controlled experiment was conducted to enhance the thermal removal of chlorine in samples prepared via the impregnation method: For samples of 2 wt% Pt, while some were prepared using the typical impregnation method with sintering at 825 °C for 2 h in air, others were prepared using a modified impregnation method. For these samples, $SnO_2$ nanoparticles and 0.1 M $H_2PtCl_6$ solution were mixed in deionized water, magnetically stirred, dried, and then the dried mixed powder was heat-treated at 825 °C for 2 h in air. Such a heat treatment on mixed powder should be more effective for removing chlorine than on pressed pellets. After the heat treatment, pellets were pressed from the powder and were sintered at 825 °C for 2 h in air. A calcination process had been added, which is termed the impregnation and calcination method hereafter. Figure 6 shows the room-temperature hydrogen sensing characteristics for two separate samples prepared using these two methods. First of all, it should be pointed out that the sample of 2 wt% Pt prepared using the impregnation method only showed a room-temperature response value of 5 to 1% $H_2$–20% $O_2$-$N_2$, which was much smaller than that of the sample of 0.5 wt% Pt prepared using the same method in Figure 5. Pt content has been found to have a significant influence on the room-temperature hydrogen sensing characteristics of Pt-$SnO_2$ composite nanoceramics [31], and 0.5 wt% Pt was found as an attractive Pt content for samples prepared using the impregnation method in this study. More interestingly, it can be seen that the sample prepared using the impregnation and calcination method exhibited much better room-temperature hydrogen sensing characteristics than the other one in Figure 6, which showed a room-temperature response value of 2750 to 1% $H_2$–20% $O_2$-$N_2$. The calcination process of 825 °C for 2 h in air dramatically affected those samples prepared using the impregnation method. In other words, for samples prepared using the impregnation method, much longer time durations at high temperatures are needed to optimize the catalytic effect of noble metals than those prepared using the solution reduction method. For the preparation of many noble metal-loaded nanomaterials, such long-time durations at high temperatures are often not practicable. This result further confirms the importance of the solution reduction method in loading noble metals as nano-catalysts.

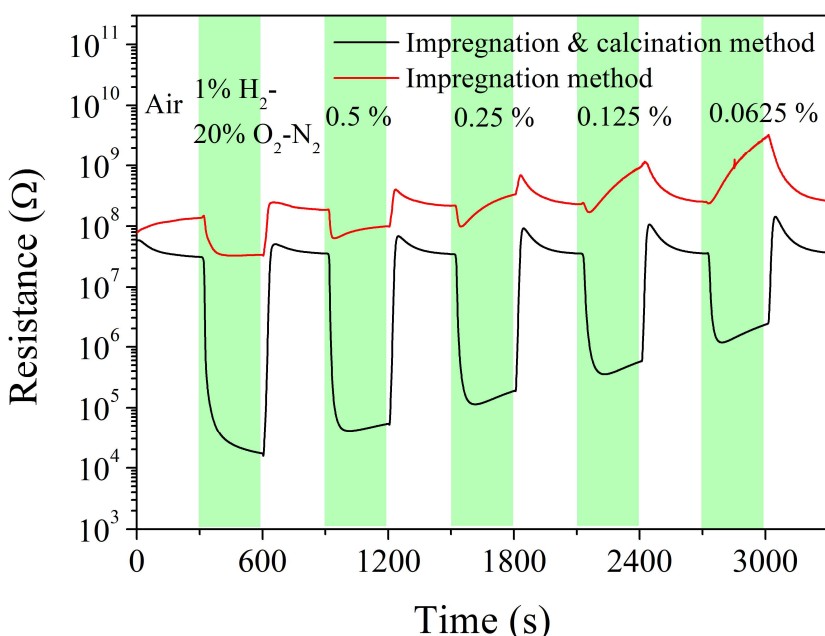

**Figure 6.** Room-temperature hydrogen sensing characteristics for two Pt-SnO$_2$ composite nanoceramic samples of 2 wt% Pt and sintered at 825 °C, prepared using the impregnation method and the impregnation and calcination method for Pt loading separately.

Furthermore, it should be emphasized that chlorine is removed at room temperature in the solution reduction method, which also forms a sharp contrast with thermal removal. According to Equation (3), for those samples prepared using the impregnation method, Cl$_2$ will react with them at elevated temperatures. It is well known that as a room-temperature hydrogen-sensitive material, Pt-SnO$_2$ composite nanoceramics can be easily affected by harmful gases [2]. The reaction of Cl$_2$ at elevated temperatures will inevitably show some negative effect on the room-temperature hydrogen sensing capability of Pt-SnO$_2$ composite nanoceramics. In short, the solution reduction method is better than the impregnation method to prepare room-temperature hydrogen Pt-SnO$_2$ composite nanoceramics in two ways: chlorine can be more thoroughly removed, and the reaction of Cl$_2$ at elevated temperatures can be mostly avoided. With these advantages, the striking difference in room-temperature hydrogen sensing characteristics between samples prepared using these two different Pt loading methods separately can now be well understood. Cl$_2$ influences many materials negatively and should be removed at as the lowest possible temperature.

In many applications, soluble noble metal salts are widely used to load noble metals as nano-catalysts on other materials [42–46]. Though an apparent catalytic effect was often observed, there was little chance to study if an optimized process had been adopted to load the noble metals. Now, through studying the room-temperature hydrogen sensing characteristics of Pt-SnO$_2$ composite nanoceramics prepared using two Pt loading methods (the impregnation and the solution reduction methods) separately, it is clearly shown that a striking difference can be observed in the catalytic effect achieved among different loading methods, even when the same noble metal salt is adopted. This is reasonable because catalysts are generally very vulnerable in many applications [47–49]. Much more attention should be directed to optimizing the catalytic effect of noble metal nano-catalysts when noble metal salts are used to load them in future research.

## 3. Materials and Methods

Pt-SnO$_2$ composite nanoceramics were prepared from SnO$_2$ nanoparticles (50–70 nm, Shanghai Aladdin Biochemical Technology Co., Shanghai, China) and H$_2$PtCl$_6$·6H$_2$O (Pt 37.5%, Shanghai Aladdin Biochemical Technology Co.) using two different Pt loading methods: (1) For the solution reduction method, SnO$_2$ nanoparticles and zinc powder

(Zn 95.0%, Sinopharm Chemical Reagent Co., Shanghai, China) were dispersed at a series of designed ratios in deionized water separately and magnetically stirred. For every suspension, 0.1 M $H_2PtCl_6$ solution (prepared from $H_2PtCl_6 \cdot 6H_2O$) was slowly dropped to react with the zinc powder. After the reaction, the suspensions were centrifuged, dried, and pressed into pellets 10 mm in diameter and 1.2 mm thick. (2) For the impregnation method, $SnO_2$ nanoparticles and 0.1 M $H_2PtCl_6$ solution were mixed in deionized water at a series of molar ratios. After being magnetically stirred for 30 min, the suspensions were dried in an oven at 110 °C for 10 h and pressed into pellets 10 mm diameter and 1.2 mm thick.

The pressed pellets were sintered at designated temperatures in air for 2 h separately. The heating rate was 3 °C/min, and after 2 h of dwelling at a designated temperature, the furnace was cooled to room temperature naturally through turning the power off. After sintering, a pair of parallel In-Ga electrodes, with 2 mm, was formed on a major surface of some sintered pellets for subsequent gas sensing measurement.

The responses of the samples to hydrogen were measured in terms of their resistance through a commercial gas sensing measurement system (GRMS-215, Partulab Com., Wuhan, China) [2], in which a constant DC voltage was applied between the In-Ga electrodes of the samples and the flown currents were measured to yield the resistances. For the response process, a target gas prepared from $O_2$, $N_2$, and 1.5% H2–N2 was flown into the measurement chamber at the rate of 300 mL/min. For the recovery process, ambient air was pumped into the chamber at the rate of 1000 mL/min. The room temperature was maintained at 25 °C, and the relative humidity (RH) in the air was kept at around 50% for the measurements. The humidity in the air was adjusted through a commercial humidifier.

The crystal structure of the Pt-$SnO_2$ composite nanoceramics was characterized through X-ray diffraction (XRD; SmartLab, Japan Rigaku, Cu K$\alpha$ radiation). The microstructure of Pt-$SnO_2$ was observed through field emission scanning electron microscopy (FESEM; SIGMA, ZEISS Corporation, Jena, Germany) and transmission electron microscopy (TEM, JEM-2100F, JEOL, Tokyo, Japan). Pt loading was analyzed using EDS (EDS; ULTIM MAX, Oxford Instruments) and HRTEM. The materials were entrusted to Shiyanjia lab (www.shiyanjia.com, accessed on 3 April 2023.) to measure the amount of residual chlorine in them according to national standard GB/T 15453-2018.

## 4. Conclusions

Pt-$SnO_2$ composite nanoceramics were prepared separately from $SnO_2$ nanoparticles and $H_2PtCl_6 \cdot 6H_2O$ using two Pt loading methods, the solution reduction method and the impregnation method. Though strong room-temperature responses to hydrogen can be observed for some samples prepared using these two Pt loading methods separately, samples prepared using the solution reduction method show much better room-temperature hydrogen sensing characteristics than those prepared using the other method. For two samples of 0.5 wt% Pt and sintered at 825 °C, the one prepared using the impregnation method had a response of 145 to 1% H2–20% $O_2$-$N_2$ at room temperature. In contrast, the other one prepared using the solution reduction method had a response of 9700. XRD, FESEM, and TEM analyses revealed no detectable difference between samples prepared using these two Pt loading methods separately, while samples prepared using the impregnation method were found to have more chlorine than those prepared using the other method. It is proposed that the difference in the chlorine removal process between these two Pt loading methods has resulted in the observed different room-temperature hydrogen sensing characteristics between samples prepared using these two methods separately. This study highlights the importance of choosing a proper loading method when noble metals are loaded from their soluble salts as nano-catalysts in many applications.

**Author Contributions:** Conceptualization, J.Z. and W.C.; methodology, J.Z., J.S. and X.L.; validation, Z.Y., F.C. and W.C.; formal analysis, J.Z. and M.W.; investigation, J.Z., J.S., Z.Y. and F.C.; data curation, J.Z., X.L. and M.W.; writing—original draft preparation, J.Z., Z.Y., F.C. and W.C.; writing—review and editing, J.Z. and W.C.; funding acquisition, J.Z. and W.C. All authors have read and agreed to the published version of the manuscript.

**Funding:** This research was funded by the Scientific Research Project of Jilin Provincial Department of Education under Grant No. JJKH20230915KJ, the Jilin Province Science and Technology Development Program under Grant No. YDZJ202201ZYTS548, and the National Key R&D Program of China under Grant No. 2020YFB2008800.

**Data Availability Statement:** The data that are unavailable in this article will be shared on reasonable request to the corresponding author.

**Conflicts of Interest:** The authors declare no conflict of interest.

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
