# Peer review of "Room-Temperature Hydrogen-Sensitive Pt-SnO2 Composite Nanoceramics: Contrasting Roles of Pt Nano-Catalysts Loaded via Two Different Methods"

_inorganics, doi:10.3390/inorganics11090366_

Round 1
Reviewer 1 Report
The article leaves an ambiguous impression. On the one hand, good sensory characteristics have been achieved. The room temperature responses are quite large and the relaxation times are low. The merits of the work should also include the qualitative characterization of the obtained materials. The submitted materials are worthy of publication.
On the other hand, there are also disadvantages in the work. The main thesis of the article (as I understand it) is as follows: of the two methods of doping SnO2 with platinum, one should choose the one that leaves the minimum amount of chlorine in the resulting gas-sensitive material. This is indicated by Table 1. But in this case, the choice of H2PtCl6 as a dopant is not clear, since there are platinum salts that do not contain chlorine, for example, Pt(NH3)2(NO2)2.
The second drawback of the article is the strange form of presentation of the material. Usually, in the Introduction, authors provide a review of the literature, then Materials and Methods, and then Results and Discussion. In this article, after the Introduction, there is not Materials and Methods, but Results and Discussion, and for some reason literary data got into this part of the article.
I believe that the article can be published after editing.
Reviewer 2 Report
Introduction:
1) pag. 1, line 40: Which is the reason? Were they developed to operate in high temperatures due to the application? Or only in high temperatures does it work properly?
2) Pag. 2, 1st paragraph: The message that the authors would like to transmit is not clear. If the sensor is for H2, one could give more examples of H2 sensors or if it is a novelty and there are no other H2 sensors like this, one can keep it clear that is a novelty.
3) The introduction could be restructured to be more clear.
Results and discussion
1) Pag. 3, line 100: Is there any reason for those temperatures? Crystal phase? From literature? One could make the reason for the temperatures more clear and if needed, add a reference.
2) Pag. 4, line 105: Is it bad or good? Which information can we get from it?
3) Pag. 4, lines 115 to 127: I would recommend some small conclusions and discussion about the quantitative values extracted from the plots. For example, 100 nm is a good size for Pt NPs for your application? Or is it only a piece of information without any connection?
4) The authors could add one table with the comparations between the systems.
5) The reason behind the change in the Pt loading is not clear.
6) One analytical curve (resistance vs concentration) could be added to Figures 4, 5, and 6. Besides the linear equation and quantitative information as the limit of detection and sensitivity.
7) Being a sensor for gas, the information about the velocity of response could be added and discussed. If one explosive gas needs to be detected, it needs to be done as fast as possible.
8) Figure 5: The resistance of the impregnation method in the absence of H2 is higher than the solution reduction method being clear that there is a difference in the material resistance. Is there any reasonable reason for it?
